# RBM-Based Simulated Quantum Annealing for Graph Isomorphism Problems

## Abstract

The graph isomorphism problem remains a fundamental challenge in computer science, driving the search for efficient decision algorithms. Due to its ambiguous computational complexity, heuristic approaches such as simulated annealing are frequently used, achieving high solution probabilities while avoiding exhaustive enumeration. However, traditional simulated annealing usually struggles with low sampling efficiency and reduced solution-finding probability in complex or large graph problems. In this study, we integrate the principles of quantum technology to address the graph isomorphism problem. By mapping the solution space to a quantum many-body system, we developed a parameterized model for variational simulated annealing. This approach emphasizes the regions of the solution space that are most likely to contain the optimal solution, thereby enhancing the search accuracy. Artificial neural networks were utilized to parameterize the quantum many-body system, leveraging their capacity for efficient function approximation to perform accurate sampling in the intricate energy landscapes of graph problems.

## 1 Introduction

Graph isomorphism, which classifies graphs by structural equivalence, originated from chemical structure comparisons in the 1950sRay & Kirsch (1957) and remains a foundational challenge in theoretical computer science. Although widely applied in pattern recognitionRiesen (2015), biochemistryMerkys et al. (2023), and communication securityLestringant et al. (2015), it is still unknown whether the problem belongs to P or is NP-complete. Despite Babai's quasi-polynomial time algorithmBabai (2016) marking significant progress, the computational complexity continues to motivate new approaches.

This complexity has spurred the development of numerous heuristic classical algorithms, such as backtracking searchConte et al. (2004), recursive partitioningMessmer & Bunke (1998), NautyMcKay et al. (1981), LADSolnon (2010), and the VF family of algorithmsCordella et al. (2004; 2001). These often employ pruning strategies based on vertex degrees, neighborhoods, or subgraph features to reduce the search space. Among these, the Simulated Annealing Algorithm (SA) has emerged as a notable method inspired by thermodynamic processesZeguendry et al. (2023); Onizawa et al. (2022); Delahaye et al. (2019); however, its efficiency is often hindered by slow sampling in rugged optimization landscapes and multiple local extrema, underscoring the ongoing need for novel algorithms. Quantum computing offers promising alternatives, including quantum walks for efficient state-based matchingTamascelli & Zanetti (2014); Rudinger et al. (2012); Wang et al. (2018); Li et al. (2023), quantum annealing formulated as QUBO/Ising modelsde Falco et al. (1988); Apolloni et al. (1989); Warren (2013); Minamisawa et al. (2019) and implemented on devices like D-WaveWang et al. (2016); Gaitan & Clark (2014); Borowski et al. (2020), and quantum methods that test isomorphism via graph invariantsMills et al. (2019); Brádler et al. (2021); Qiang et al. (2021). Nevertheless, current quantum approaches face challenges in the NISQ era, such as qubit limitations, noise susceptibility. Additionally, quantum walks are hindered by the regularity of the graph structure, particularly for strongly regular graphs; quantum annealing requires strict adherence to the adiabatic condition, which is difficult to achieve for large-scale graphs due to complex Hamiltonian structures and small energy gaps. Although NISQ devices hinder the full potential of quantum computing Di Meglio et al. (2024), they can still enhance classical algorithms by simulating quantum principles. This has led to the rise of hybrid quantum-classical algorithmsLi & Pan (2023); Crosson & Harrow (2016); Bando & Nishimori (2021); Zeng et al. (2024), integrating clas-

sical machine learning models with quantum computing, which has become a key area of research. One notable approach is Simulated Quantum Annealing (SQA)Crosson & Harrow (2016); Bando & Nishimori (2021), a quantum-inspired classical algorithm that employs the Path Integral Monte Carlo (PIMC) methodIsakov et al. (2016) to map the quantum Hamiltonian onto a classical system via Trotter decomposition (forming "Trotter slices"). By simulating quantum tunneling effects, SQA offers a higher probability to escape local optima compared to SA or classical heuristics.

This work introduces an SQA algorithm built on a neural-network framework. By employing a Restricted Boltzmann Machine to parameterize the quantum system, we recast the graph isomorphism problem as a quantum many-body challenge and solve it using variational neural-network quantum states (NQS). NQS has garnered significant interest in quantum mechanics for solving problems such as ground statesCarleo & Troyer (2017), dynamic evolutionNomura et al. (2017), and quantum tomographyCarleo et al. (2019). Unlike existing PIMC-based SQA methods, our approach constructs an RBM-based effective Hamiltonian that captures the system's energy landscape. In principle, by dynamically optimizing a small sampling space, our method increases the likelihood of identifying the global optimum. Moreover, rapid variational optimization can potentially reduce the computational time per iteration, making our approach attractive for large-scale graph problems. The paper is organized as follows: Section 2 formalizes the graph isomorphism problem and its QUBO/Ising formulations; Section 3 details the proposed methodology; Section 4 presents experimental results and comparisons with SA and SQA; Section 5 presents the conclusions.

## 2 PRELIMINARIES

**QUBO Formulation for Graph Isomorphism Problems** We assume that the graphs under discussion are simple undirected graphs without multiple edges or self-loops unless otherwise specified. The key to solving the graph isomorphism problem is to find a mapping between the two sets of vertices that satisfies both the vertex correspondence and edge invariant constraints. Two graphs, $G_1$ and $G_2$, are isomorphic if and only if a permutation matrix $P$ can be found such that $PA_1P^{-1} = A_2$, with $A_1$ and $A_2$ representing the adjacency matrices of $G_1$ and $G_2$ respectively, as shown in Fig.1.

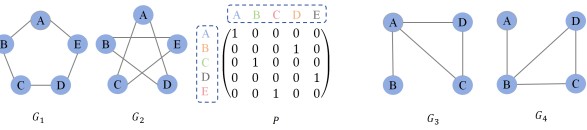

Figure 1: $G_1$ and $G_2$ are isomorphic, while $G_3$ and $G_4$ has the same degree sequence.

Several methods for transferring isomorphic problems into QUBO problems have been discussed **?**, including the integer Programming (IP) formulation, and the direct method, which is a one-dimensional representation of the permutation matrix $P$, denoted as, $\vec{x} = [x_{AA}, x_{AB}, x_{AC}, x_{AD}, x_{AE}, x_{BA}, x_{BB}, x_{BC}, x_{BD}, x_{BE}, x_{CA}, x_{CB}, x_{CC}, x_{CD}, x_{CE}, x_{DA}, x_{DB}, x_{DC}, x_{DD}, x_{DE}, x_{EA}, x_{EB}, x_{EC}, x_{ED}, x_{EE}]$ where each component $x_{ij}(i \in G_1, j \in G_2)$ is either 0 or 1,depending on the mapping relationship. The experiment on the D-Wave architecture suggests that the direct formulation is more practical. However, we find that, in practice, many graph isomorphism instances involve non-regular graphs, allowing for further optimization of the one-dimensional solution vector $\vec{x}$. Since graphs with different degree sequences cannot be isomorphic, and even with identical degree sequences, vertices can only map to others of the same degree, this enables a reduction in the solution space. To illustrate, consider Fig.1, where graphs $G_3$ and $G_4$ have the same degree sequence $\{1, 2, 3, 3\}$. The vertex set is partitioned into subsets by degree, restricting possible mappings: $B$ to $A$, $\{A, C\}$ to $\{B, D\}$, $D$ to $C$. This reduces the solution vector length from 16 to 6: $\vec{x} = [x_{AB}, x_{AD}, x_{BA}, x_{CB}, x_{CD}, x_{DC}]$. This optimization significantly reduces the solution space and makes the isomorphism problem more tractable.

After encoding the solution space into the vector $\vec{x}$, the graph isomorphism problem can then be reformulated as a QUBO problem by defining an objective function $F(\vec{x})$ to be minimized:

$$\min_{x_i \in \{0,1\}} F(\vec{x}) = \vec{x}^T Q \vec{x}, \tag{1}$$

where $Q$ is a symmetric matrix. The matrix element $Q_{ii}$ represents the linear coefficient of a single variable in $\vec{x}$, while $Q_{ij}$ represents the quadratic interaction coefficient between two variables. $\vec{x} = \{x_{ij}\}^{|S|}$ is a binary solution vector of length $|S|$ and $x_{ij} \in \{0, 1\}$ represents the mapping relationship between vertices $i \in V_1$ and $j \in V_2$. Let $S = \{(i, j)|i \in V_1, j \in V_2\}$ be the set of possible vertex mapping pairs, considering the consistency of the node degrees between $V_1$ and $V_2$. The objective function comprises two parts:

$$F(\vec{x}) = F_1(\vec{x}) + F_2(\vec{x}), \tag{2}$$

where

$$F_1(\vec{x}) = \sum_{i \in V_1} (\sum_{(i,j) \in S} x_{ij} - 1)^2 + \sum_{i \in V_2} (\sum_{(i,j) \in S} x_{ij} - 1)^2, \tag{3}$$

and

$$F_2(\vec{x}) = \sum_{(i,j) \in S} \sum_{(k,l) \in S} [x_{ij} x_{kl} (A_{1(i,k)} - A_{2(j,l)})^2]. \tag{4}$$

$F_1(\vec{x})$ is used to ensure the properties of the permutation matrix corresponding to $\vec{x}$, that is, each row and each column has only one element that is $1$. This ensures compliance with the vertex correspondence constraint. $F_2(\vec{x})$ ensures that the mapping relationship complies with the second constraint. The term $(A_{1(i,k)} - A_{2(j,l)})^2$ ensures that only edges matching in both graphs contribute to minimizing $F_2(\vec{x})$. Let $\vec{x_0}$ be the optimal solution finally obtained. If $F(\vec{x_0}) = 0$, this implies that $G_1$ and $G_2$ are isomorphic. The binary entries of $\vec{x_0}$ encode the vertex mapping between $G_1$ and $G_2$ satisfying both constraints. If $F(\vec{x_0}) > 0$ the graphs are not isomorphic, as no valid mapping exists that satisfies both constraints simultaneously.

**Ising optimization problem** The Ising model originated in the study of spin systems within physics, initially developed to describe phase transitions in ferromagnetic materials Volkov & Kopaev (1974). The goal of the Ising optimization problem is to guide an $N$-qubit quantum system $|\psi\rangle$ toward the ground state $|\vec{s}\rangle = |s_0, s_1, \ldots, s_{N-1}\rangle$ of a problem Hamiltonian $\hat{H}$, such that the energy $\langle\vec{s}|\hat{H}|\vec{s}\rangle$ is minimized. This objective aligns with minimizing the QUBO objective function, with the key distinction that binary variables in QUBO correspond to projections ($\pm 1$) of single-qubit eigenstates $|+\rangle$ and $|-\rangle$ under the Pauli-Z operator. In the quantum formulation, each classical binary variable is replaced by the Pauli-Z operator $\sigma_i^z$, and the Hamiltonian takes the general form:

$$\hat{H} = -\sum_{i,j=1}^{N} J_{ij} \sigma_i^z \sigma_j^z - \sum_{i=1}^{N} h_i \sigma_i^z, \tag{5}$$

where $J_{ij}$ is the coupling constant describing the spin pair $\sigma_i^z, \sigma_j^z$, and $h_i$ is the effect of the external magnetic field on each spin. Currently, multiple quantum approaches for solving Ising problems can be found in the literature, including quantum annealingWarren (2013); Minamisawa et al. (2019), variational quantum algorithms (VQAs), and the quantum approximate optimization algorithm (QAOA)Farhi et al. (2014); Li et al. (2025); Wang et al. (2025). Given the limitations of the NISQ era, this work adopts a classical computational framework that incorporates simulations of quantum algorithms to retain certain quantum advantages.

## 3 RBM-BASED SQA FOR GRAPH ISOMORPHISM

**Hamiltonian Construction of Graph Isomorphism Problem** For a graph isomorphism instance with a solution vector $\vec{x}$ of length $L$, $L$ spin/qubits are used for encoding. The binary variable$x_{ij}$is mapped to a spin/qubit $s_{ij}$ via the Pauli-Z operator. The solution space of the original problem is then mapped to a quantum system$|\psi\rangle$ of size $2^L$, where $(i, j) \in S$, the vertex mapping pair set. The ground state of the Hamiltonian $\hat{H}$ encodes the vertex mapping, and its energy $\lambda$ serves as a key indicator of mapping correctness. We have:

$$\hat{H} = \hat{H}_1 + \hat{H}_2, \tag{6}$$

where

$$\hat{H}_1 = \sum_{i=1}^{N} (\sum_{(i,j) \in S} \frac{\sigma_{ij}^z + I}{2} - I)^2 + \sum_{j=1}^{N} (\sum_{(i,j) \in S} \frac{\sigma_{ij}^z + I}{2} - I)^2, \tag{7}$$

and

$$\hat{H}_2 = \sum_{(i,j)\in S} \sum_{(k,l)\in S} \frac{1}{4}[(\sigma_{ij}^z \sigma_{kl}^z + \sigma_{ij}^z + \sigma_{kl}^z + I)(A_{1(i,k)} - A_{2(j,l)})^2]. \tag{8}$$

The eigenstate of $\hat{H}$ correspond to the different configurations of $x_{ij}$, where each configuration (with $x_{ij} \in \{0,1\}$) represents a possible mapping relationship in the graph isomorphism problem, and is denoted as $|\vec{s}\rangle$. For any eigenstate $|\vec{s}\rangle$ of $\hat{H}$, which serves as the quantum representation of a mapping solution to the graph isomorphism problem, the associated energy eigenvalue is defined as,

$$\langle \vec{s}|\hat{H}|\vec{s}\rangle = \begin{cases} 0, & \text{the right mapping combination} \\ \lambda, & \text{otherwise} \end{cases} \tag{9}$$

Here, $\lambda$ quantifies the number of constraint violations in the graph isomorphism mapping, as defined by equations equation 3 and equation 4. Initially, the bijection-related objective function in equation 3 is evaluated. Notably, any redundant mapping increases the value of $F_1(\vec{x})$ by 1. In order to determine the upper bound of $F_1(\vec{x})$, it is essential to include the maximum possible number of redundant mappings beyond the valid bijection that satisfies the cost function. Consider two simple undirected graphs, $G_1$ and $G_2$ each with $N$ vertices. Assume that every vertex in $G_1$ maps to every vertex in $G_2$, and each vertex has one correct mapping among the $N$ mapping relationships, with the remaining mappings being redundant. According to equation 3, this results in a penalty value of $2N(N-1)^2$. It is important to note that when traversing the vertices of $G_1$ and $G_2$ accounts for $F_1(\vec{x})$ each redundant mapping twice. Therefore, the upper bound of $F_1(\vec{x})$ is given by $2N(N-1)^2$.

Then, we check the edges mapping objective function $F_2(\vec{x})$. Under the condition that $F_1(\vec{x})$ reaches its maximum value, the length of $\vec{x}$ is $N^2$, implying that both graphs $G_1$ and $G_2$ are fully connected. As a result, $x_{ij} = x_{kl} = 1, \forall (i,j), (k,l) \in S$. For the term $(A_{1(i,k)} - A_{2(j,l)})^2$, the following cases are considered:

$$(A_{1(ik)} - A_{2(jl)})^2 = \begin{cases} 0, & \text{if } i \neq k \text{ and } j \neq l, \\ 0, & \text{if } i = k \text{ and } j = l, \\ 1, & \text{others}, \end{cases} \tag{10}$$

when $i \neq k$ and $j \neq l$, means under the mapping $x_{ij} = x_{kl} = 1$, the edge invariant constraint. In the case $i = k$ and $j = l$, there is $A_{1(i,k)} = A_{2(j,l)} = 0$. Since there is no cycle in a simple undirected graph. The third case, where $i = k$ and $j \neq l$ or $i \neq k$ and $j = l$, the absence of cycles in the graph ensures that $A_{1(i,k)}$ and $A_{2(j,l)}$ are not simultaneously zero, resulting in the theoretical maximum value $2N^2(N-1)$. $F_2(\vec{x})$ checks whether the edge relationships are consistent for mapped vertex pairs, incrementing its value by 1 for each violation. However, it does not verify compliance with bijection principles, as this is constrained by $F_1(\vec{x})$. Since $F_1$ and $F_2$ capture disjoint constraints, the upper bound of $\lambda$ is the sum of their maxima: $\lambda \leq 2N^2(N-1) + 2N(N-1)^2$.

**Neural Network Quantum States Approximate Wavefunctions** To represent the solution space, we employ Neural-network Quantum States (NQS), specifically a Restricted Boltzmann Machine (RBM), to efficiently represent the complex wave function of the quantum system encoding graph isomorphism solutions. This approach is particularly relevant to quantum many-body problem, where wave functions of interacting particles exhibit exponentially scaling complexity, making analytical treatments infeasible Verstraete (2015). This complexity renders such systems analytically intractable. Building on the foundational work of Carleo & Troyer (2017), which demonstrated the strong expressiveness of neural networks for approximating many-body wave functions, we apply variational Monte Carlo (VMC) to estimate the ground state energy of $\hat{H}$ with high precision. This combined NQS-VMC framework leverages machine learning advances that have significantly impacted research in condensed matter physics Jia et al. (2019); Gao & Duan (2017).

RBM is a two-layer bipartite graph, typically consisting of two types of units: hidden layer units and visible layer units, as shown in Fig.2. RBM uses hidden units to model high-order and non-linear patterns in data, enhancing the capability to represent wave functions Torlai & Melko (2018). For a solution vector $\vec{x}$ of length $L$ in the graph isomorphic problem, the corresponding quantum eigenstate is defined as $|\vec{s}\rangle = |s_i\rangle^{\otimes L}$, which is used as input to interact with the bias vector $\vec{a} = \{a_i\}$ in the visible layer. This input is fully connected to the hidden layer containing $H$ binary hidden

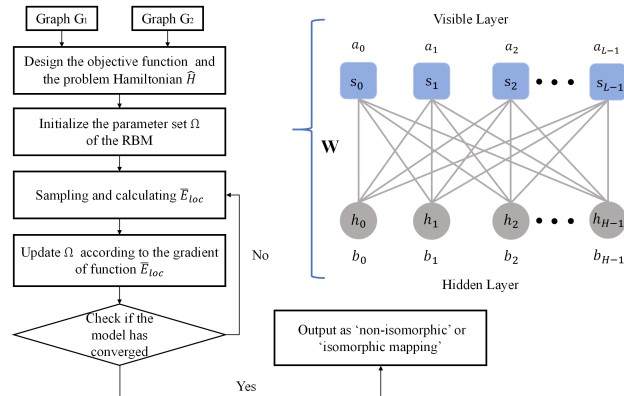

Figure 2: Proposed RBM-based GI algorithm and the structure of a Restricted Boltzmann Machine.

nodes. The bias vector of the hidden layer is $\vec{b} = \{b_j\}$,where $s_i, h_j \in \{1, -1\}$. The connection relationship between each vertex in the two layers is determined by the corresponding connection weight parameter $W = \{W_{ij}\}$. The complete set of parameters for the RBM model is given by $\Omega = \{\vec{a}, \vec{b}, W\}$, where $\Omega_i \in R$. The wave function of the quantum system can be expressed as:

$$\psi(\vec{s}, \Omega) = \sum_{\vec{h}} e^{\sum_i s_i a_i + \sum_{ij} s_i W_{ij} h_j + \sum_j h_j b_j} \tag{11}$$

Given that the units in the RBM layers are not connected within their respective layers, we simplify:

$$\psi(\vec{s}, \Omega) = e^{\sum_i a_i s_i} \prod_j 2 \cosh(b_j + \sum_i W_{ij} s_i) \tag{12}$$

Therefore, after mapping the solution space of the graph isomorphism problem to the quantum system, the corresponding neural network quantum state can be written as:

$$|\psi\rangle = \sum_{\vec{s}} \psi(\vec{s}, \Omega)|\vec{s}\rangle \tag{13}$$

Now, to calculate the expectation value of the Hamiltonian $\hat{H}$,we use,

$$\langle\psi(\vec{s}, \Omega)|\hat{H}|\psi(\vec{s}, \Omega)\rangle = |\psi(\vec{s}, \Omega)|^2 \lambda = P(\vec{s})\lambda \tag{14}$$

Here,$|\psi(\vec{s}, \Omega)\rangle = \psi(\vec{s}, \Omega)|\vec{s}\rangle$ used to represent the state of non-normalized mapping $|\vec{s}\rangle$. Therefore, we realize the parameterized representation of the quantum system through ansatz in equation 12. In quantum mechanics, the eigenstate wave function in the system determines the probability that the eigenstate is observed to some extent, as shown in equation 13. This provides a theoretical foundation for mapping quantum systems into probability distribution space.

By adjusting the energy of the quantum system through Monte Carlo sampling and applying the variational principle, we aim to approximate the ground state and minimize the energy of the system. In detail, for the problem Hamiltonian $\hat{H}$ of the graph isomorphism problem, the energy of the system represented by the NQS state $|\psi\rangle$ is given by:

$$E_{global} = \frac{\langle\psi|\hat{H}|\psi\rangle}{\langle\psi|\psi\rangle} \tag{15}$$

This represents the expectation value of the Hamiltonian $\hat{H}$ with respect to the NQS $|\psi\rangle$, providing a measure of the system's energy. To estimate this, we collect $n$ samples $|\vec{s}\rangle^n$—eigenstates of $\hat{H}$—using the Metropolis-Hastings algorithm. This algorithm constructs a Markov chain to generate samples from the target distribution $P(\vec{s})$, corresponding to the squared modulus of the wave function, and accepts or rejects proposed states based on their relative probabilities.

$$E_{loc} = \sum_{\vec{s}} \langle\psi(\vec{s}, \Omega)|\hat{H}|\psi(\vec{s}, \Omega)\rangle = \sum_{\vec{s}} P(\vec{s})\lambda \tag{16}$$

$$\bar{E}_{loc} = \frac{1}{M} \sum^{M} E_{loc} \approx E_{global} \tag{17}$$

The local energy $E_{loc}$ is calculated from $M$ sample configurations, and the system's overall energy expectation is estimated by the sample mean $\bar{E}_{loc}$. By mapping the graph isomorphism problem to finding the ground state of the quantum Hamiltonian $\hat{H}$, we aim to minimize the global energy $E_{global}$ by continuously adjusting the parameters of the variational wave function parameters $\Omega = \{\vec{a}, \vec{b}, W\}$, gradually increasing the probability amplitude of the eigenstate $|\vec{s}\rangle$ encoding the target solution. As the solution space increases, the Hilbert space dimension of the quantum system becomes very large. The sampling component of the VMC algorithm helps mitigate this issue by leveraging sampling techniques based on relevant statistical principles Gros (1989). Sampling introduces statistical uncertainty into the algorithm, and according to the zero-variance principle of VMC, when the system's wave function $|\psi\rangle$ approaches the true ground-state wave function, the expected variance of the sample energy tends to 0. In this case, $\bar{E}_{loc} = E_{global} = 0$.

Variance thus serves as a convergence indicator: zero variance signifies coinciding local energies and stability, while a stationary energy expectation implies vanishing parameter updates. At this stage, the sample mean approximates the ground-state energy, the target solution dominates the probability distribution, and the hit rate is maximized. Variance reduction and energy minimization jointly guide wavefunction updates, improving the accuracy of the ground-state estimate. The overall process is illustrated in Fig. 2, with an RBM-SQA example for $G_3$ and $G_4$ given in appendix (A.1).

## 4    NUMERICAL VALIDATION

In this section, we evaluate the performance of RBM-SQA against classical simulated annealing (SA) and simulated quantum annealing (SQA), both in solution accuracy and runtime when solving the graph isomorphism problem. SA is a classical probabilistic optimization technique inspired by thermal annealing. The search process is governed by the Metropolis criterion under a gradually decreasing temperature schedule. At high temperatures, the system has a probability of accepting suboptimal solutions, which in principle helps escape from local minima rather than the other classical heuristic methods. SQA is a quantum-inspired algorithm that uses path-integral Monte Carlo (PIMC) and Trotter slicing to emulate quantum tunneling effects, here we call PIMC-SQA. By mapping the quantum system to a classical system, SQA enable more effective traversal of energy barriers compared to SA. The specifics of SA and PIMC-SQA can be found in the appendix (A.2).

For fair comparison, SA PIMC-SQA and RBM-SQA are configured with the same number of total solution space explorations, set as $N_{sweep} \times N_{annealing}$, where $N_{annealing} = 1000$ and $N_{sweep}$ scales with graph size. In SA, the temperature decreases from $T_0 = 100$ to $T_{final} = 0.667$; in SQA, the transverse field decreases from $\Gamma_0 = 100$ to $\Gamma_{final} = 0.667$ with four Trotter slices; and in RBM-SQA, the sampling scale per iteration is $N_{sweep}$ with a learning rate in $[0.01, 0.1]$. This consistent setup allows us to isolate the intrinsic efficiency and accuracy of each algorithm.

**Isomorphic Graph** We tested the RBM-SQA on isomorphic graph pairs with $N = 4, 5, 6, 7, 8, 9, 10$ vertices. Across all instances, the algorithm consistently achieved a hit rate of 1, demonstrating reliable convergence to the ground state. The solution vector length, obtained via a vertex-mapping encoding scheme, directly specifies the qubit count and solution space dimension. Runtime per iteration also scaled with problem size, from about 0.004,s for small graphs ($N = 4$) to 0.23,s for the largest tested case ($N = 10$ with 52 qubits). These results, summarized in Table A.1, confirm that while computational cost grows with the encoding dimension, the RBM-based SQA maintains stable accuracy across different graph sizes.

To further examine the behavior of the RBM-based SQA, we selected a representative isomorphic graph pair for each $N = 5, 6, 7, 8, 9, 10$ and compared the results with classical SA and PIMC-based SQA. The performance on each graph instance is shown in Fig. 3, while Table 1 summarizes the corresponding degree sequences, solution space dimensions, qubit counts, and per-iteration sample sizes. Experimental results show that the RBM-SQA achieves a faster reduction in system energy, converging more quickly to the theoretical minimum. While SA and PIMC-SQA algorithms control the search direction by lowering the system temperature and transverse field strength, the RBM-SQA accounts for the overall distribution of the system. This enables the RBM-SQA to make more precise adjustments, resulting in a stable and consistent decrease in energy, ultimately maintaining

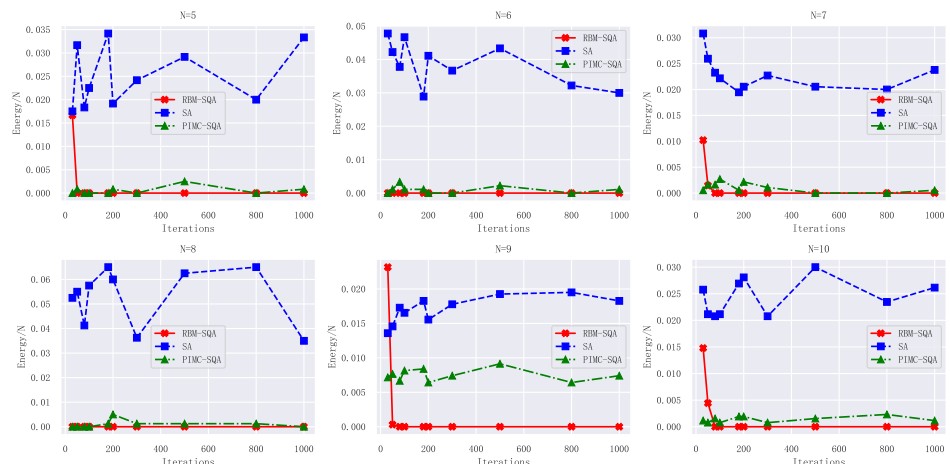

Figure 3: Average residual energy per site Energy/N of graph instances evolves over iterations.

Table 1: The settings of isomorphic graph parameters.

| $N$ | Degree Sequence | Space | Qubits | Sampling | $N$ | Degree Sequence | Space | Qubits | Sampling |
|---|---|---|---|---|---|---|---|---|---|
| 5 | $[2, 2, 2, 2, 2]$ | $2^{25}$ | 25 | 250 | 6 | $[1, 2, 2, 3, 2, 2]$ | $2^{18}$ | 18 | 180 |
| 7 | $[3, 3, 3, 3, 3, 3, 6]$ | $2^{37}$ | 37 | 370 | 8 | $[5, 4, 5, 4, 3, 2, 1, 4]$ | $2^{16}$ | 16 | 160 |
| 9 | $[8, 8, 8, 8, 8, 8, 8, 8, 8]$ | $2^{81}$ | 81 | 810 | 10 | $[7, 7, 8, 8, 7, 8, 8, 8, 8]$ | $2^{52}$ | 52 | 520 |

the theoretical global minimum. Although RBM-SQA often starts with higher initial energy values due to stochastic initialization and M-H batch sampling, this does not hinder convergence; instead, it accelerates the overall energy decline and stabilizes the system at the global minimum.

Sampling size, treated as a key hyperparameter, directly influences convergence speed and accuracy. Fig.4 shows the effect of sampling scale on model performance. Due to hardware constraints, in this paper, experiments were conducted on graphs with a maximum of 18 vertices. In these experiments, setting the sampling scale to ten times the number of encoded qubits yielded the best balance between efficiency and accuracy. For larger or more complex graphs, the optimal scale may be changed based on the problem characteristics.

After training, we sample the estimated ground state and compute the hit rate by counting exact-mapping solutions. For comparison, we ran SA and PIMC-based SQA algorithms multiple times and calculated their success rates by counting the number of times the final solutions matched the target

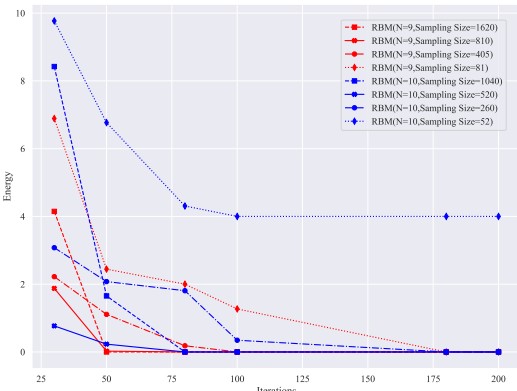

Figure 4: The influence of sampling scale: small sampling slows residual energy reduction, while high sampling increases high-energy samples, reducing early-stage ground-state hit probability.

solution. Fig.5 shows that our algorithm initially has a lower success rate, but it rapidly increases with more sampling iterations and stabilizes at 1,the overall average hit rate is 97.52%, while the SA algorithm is 66.75% and the PIMC-SQA algorithm is 94.15%. Therefore, the accuracy of this algorithm is 46.09% higher than SA and 3.58% higher than PIMC-SQA. Thus, our method, when handling rugged and glassy optimization landscapes, achieves a higher probability of finding the target solution compared to algorithms based on classical simulated annealing principles.

To complement accuracy analysis, we also compared runtime across the three algorithms. As shown in Fig. 6, SA is the fastest, averaging 0.076s per run, but with only moderate accuracy. PIMC-SQA improves accuracy but at the cost of substantially higher runtime, averaging 0.441s. The RBM-based SQA achieves the best accuracy with an average runtime of 0.161s, reducing runtime by 63.49% relative to PIMC-SQA while remaining within a practical range. Moreover, its runtime does not scale directly with the number of graph vertices, as problem encoding means that structural complexity and edge density, rather than vertex count alone, dominate convergence behavior.

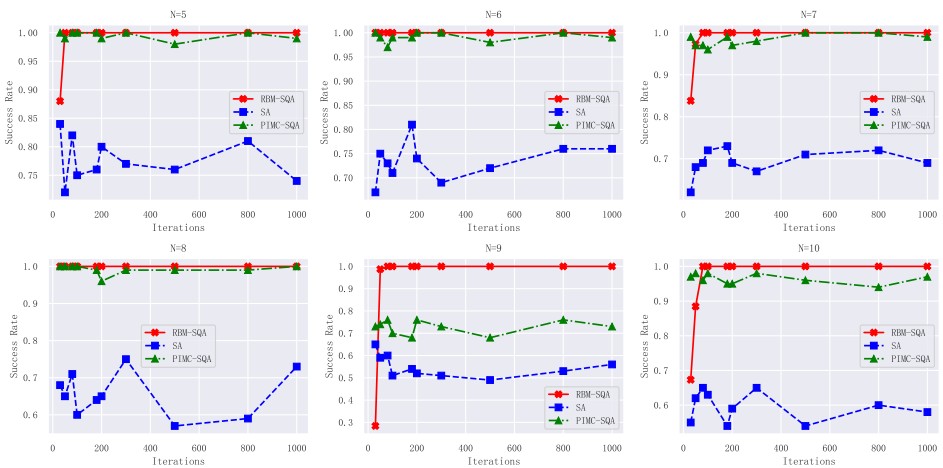

Figure 5: The hit rate of SA, PIMC-SQA, and RBM-SQA as the number of iterations changes.

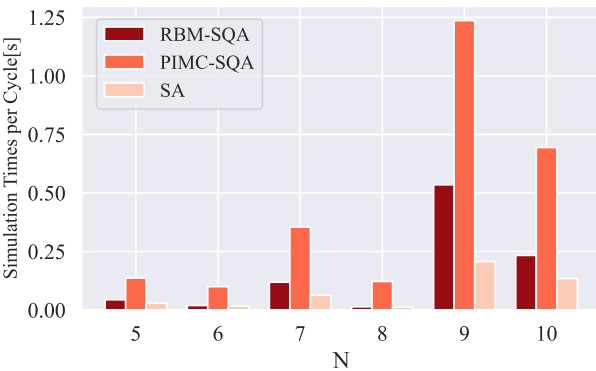

Figure 6: A comparison chart of the runtime per iteration in seconds.

**Non-isomorphic Graph** We tested non-isomorphic graphs using RBM-SQA, demonstrating its ability to distinguish them. Table 2 (additional experimental results in Table A.2) compares the performance of the RBM-based SQA, SA, and PIMC-based SQA algorithms in solving non-isomorphic instances, with the per-iteration sample size set to ten times the number of qubits (consistent with Table 1). The results show that the RBM-based SQA algorithm achieves a lower (yet nonzero) average residual energy than the other methods, demonstrating its capability in exploring global optimal vertex mappings under non-isomorphic constraints.

Table 2: The average of the final minimum energy expectations of 100 experiments on various non-isomorphic graphs.

| N | SA | PIMC-SQA | RBM-SQA | N | SA | PIMC-SQA | RBM-SQA |
|---|------|----------|---------|----|--------|----------|---------|
| 7 | 5.887 | 4.683 | 4.500 | 12 | 6.336 | 5.074 | 4.500 |
| 8 | 5.460 | 4.258 | 4.000 | 13 | 7.455 | 4.618 | 4.000 |
| 9 | 6.580 | 5.254 | 5.000 | 16 | 9.306 | 8.740 | 8.000 |
| 10 | 8.426 | 7.056 | 6.400 | 17 | 10.584 | 8.762 | 8.000 |
| 11 | 6.962 | 5.063 | 4.500 | 18 | 5.987 | 4.290 | 4.000 |

As the model parameters converge, the probability amplitude distribution of the quantum system dynamically concentrates towards the low-energy states of the Hamiltonian corresponding to the target problem. Spin configurations that align with lower energy expectations have significantly increased solution probabilities, while those that do not align see a corresponding decrease in solution probability. In particular, the system can only reach the theoretical ground state ( energy $E = 0$) when the two graphs are isomorphic. In all non-isomorphic cases, the three algorithms were ran 100 times, show an average ground state energy $> 0$, indicating the absence of valid isomorphic mappings.

A lower expected energy intuitively indicates a higher degree of optimization in the spin configuration for vertex mapping, manifesting as an increased proportion of vertex pairs in candidate solutions that satisfy isomorphism constraints. Notably, even under non-isomorphic conditions, the algorithm can still achieve maximal partial isomorphic coverage. In scenarios involving non-isomorphic graph instances, RBM-based SQA maintains the lowest average energy, outperforming both SQA and SA.

## 5 CONCLUSIONS

In this work, we employ neural-network quantum states (NQS) to represent the solution space of the graph isomorphism problem and use variational Monte Carlo (VMC) with sampling to simulate the evolution of the corresponding Ising Hamiltonian toward its ground state. The RBM-based SQA replaces the random search principle in the solution space of the SA and SQA algorithms with a variational approach. Through the parameterized optimization of the neural network, it can more precisely adjust the distribution of the solution space, thereby more effectively searching for the global optimum. Experimental results show that the average optimal solution hit rate of the proposed algorithm improves by 3.58% and 46.09% compared to the PIMC-based SQA and SA algorithms, respectively. Additionally, compared to PIMC-based SQA, our approach achieves an average reduction of 63.49% in computational time per iteration, demonstrating significant advantages in both accuracy and efficiency for solving graph-related problems.

Future research will focus on further reducing the solution space, potentially constraining it to Dicke states to exclude invalid mappings, since the quantum system corresponding to the correct solution in the problem solution space is often presented in the form of Dicke states. However, due to the characteristics of the VMC method, even an initial Dicke state may not preserve this structure during sampling updates, making it a challenge to combine Dicke constraints with variational optimization effectively. Additionally, the development of new quantum algorithms, including quantum neural networks and quantum-enhanced optimization methods, will be pursued. These efforts promise to deepen the integration of quantum computing and machine learning, enabling significant progress in solving classical NP-hard problems such as graph isomorphism.

ACKNOWLEDGMENTS

This research was supported by the National Nature Science Foundation of China (Grants No. 62101600, 62301454), State Key Lab of Processors, Institute of Computing Technology, CAS under Grant No. CLQ202404, the Beijing Natural Science Foundation (Grant No. 4252006), Fundamental Research Funds for the Central Universities (Grant No. SWU-KQ22049), and the Natural Science Foundation of Chongqing, China (Grant No. CSTB2023NSCQ-MSX0739).

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

## A  APPENDIX

### A.1  THE EXAMPLE OF RBM-SQA WITH $G_3$ AND $G_4$ IN FIG. 1

We take $G_3$ and $G_4$ in Fig. 1 as an example to check how the target distribution of samples changes with the optimization of $\Omega$. The initial parameters in $\Omega$ are randomly sampled from a normal distribution with mean 0 and standard deviation 0.01. Although the solution space is $2^6$, a small sample size here, 3 samples, is sufficient during iterative sampling. In the first iteration, the algorithm generates sample solutions: $[0, 1, 1, 0, 1, 1]$, $[1, 1, 0, 0, 1, 0]$, and $[0, 1, 1, 1, 0, 1]$, estimating a local energy $\overline{E}_{\text{loc}} = 6.25$. To steer the network toward lower-energy configurations, $\Omega$ is updated using stochastic gradient descent (SGD) combined with stochastic reconfiguration (SR), moving along the gradient direction $\nabla\Omega\overline{E}_{\text{loc}}$. This iterative process—sampling, energy estimation, and parameter update- continues over multiple steps. By the 5th iteration, sampled configurations begin clustering near the true solution. After 30 iterations, the variational wave function parameters approach convergence, the probability of the target mapping reaches 0.983. Final sampling from the converged NQS yields $\vec{x} = [1, 1, 0, 1, 1, 0]$, corresponding to the isomorphism $A \leftrightarrow B, B \leftrightarrow A, C \leftrightarrow D, D \leftrightarrow C$.

### A.2  IMPLEMENTATION DETAILS OF SA AND SQA ALGORITHMS

SA is a classical probabilistic optimization technique inspired by thermal annealing. At high temperatures, the system has a probability of accepting suboptimal solutions, which in principle helps escape from local minima rather than the other classical heuristic methods. As a typical classical heuristic algorithm, SA performance will be compared with the RBM-based algorithm in this paper. To implement the SA algorithm, we follow these steps:

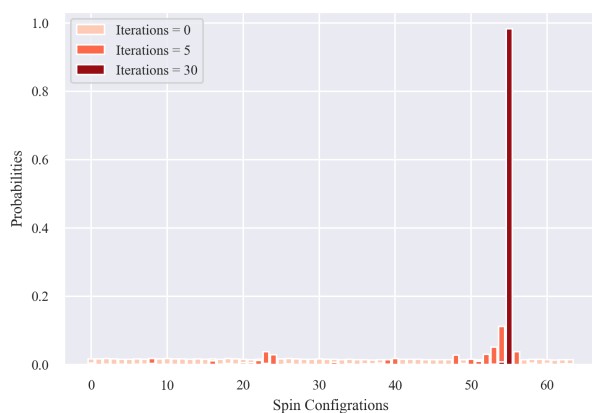

Figure 7: The sampling-optimization process for isomorphic problem of $G_3$ and $G_4$ in Fig.1.

1. Initialization. Start by initializing a solution vector $\vec{x_0}$. Set the initial temperature of the system to $T_0$ and design the objective function $F$ of the graph isomorphism problem, we will refer to it as the energy function later.

2. Exploring the Solution Space. Select a new solution $\vec{x_1}$ randomly from the neighborhood of the current solution, which is achieved by flipping a random bit in the solution vector. Compute the energy change $\delta F = F(\vec{x_0}) - F(\vec{x_1})$ and update the solution using the Metropolis criterion. If $\Delta E \leq 0$, accept the new solution; if $\Delta E > 0$, accept it with a probability of $P(\Delta E, T) = e^{-\Delta E/T}$. This probabilistic acceptance of non-optimal solutions helps escape local minima. Note that the system temperature remains unchanged in this step. We define $N_{sweep}$ as the number of solution space explorations at a fixed temperature, where $N_{sweep}$ new solution vectors are generated by flipping bits.

3. Annealing. Gradually reduce the system temperature according to the set temperature schedule, transitioning from $T_0$ to $T_1$.

4. Repeat the solution space exploration and annealing steps until the system temperature reaches the final value $T_{final}$. The number of temperature drops is denoted by $N_{annealing}$, which, together with $T_0$ and $T_{final}$, defines the annealing schedule of the SA algorithm. The process ends when the temperature reaches $T_{final}$, outputting the historical optimal solution found during the exploration.

On the other hand, SQA leverages Trotter slicing to emulate quantum tunneling, allowing it to surmount energy barriers more effectively. In true quantum annealing, adiabatic evolution of the system's Hamiltonian enables the state to "tunnel" through narrow, high barriers that would trap purely thermal algorithms. SQA partially replicates this advantage and tends to outperform SA in many scenarios, although it lacks genuine quantum coherence and entanglement. It relies on the path-integral method (PIMC) to statistically mimic quantum tunneling, based on imaginary time evolution. This approach transforms the quantum system's time-dependent evolution into a thermodynamic equilibrium problem of an equivalent classical system.

For a given Hamiltonian $H$, the imaginary time evolution operator is defined as $\widehat{U}(\beta) = e^{-\beta H}$, where $\beta = \frac{1}{T}$ is the inverse of the system temperature. For any initial state $|\psi_0\rangle$, the system can evolve toward the ground state $|\psi_f\rangle$ through imaginary time evolution, satisfying:

$$\lim_{\beta \to \infty} \hat{U}(\beta)|\psi_0\rangle = |\psi_f\rangle \tag{18}$$

Assuming the ground state energy is $E_f$, the ground state $|\psi_f\rangle$ satisfies the eigenvalue equation:

$$H|\psi_f\rangle = E_f|\psi_f\rangle \tag{19}$$

According to the theory of thermodynamic equilibrium, the probability of a quantum state under temperature $T$ satisfies:

$$P(|\psi\rangle, T) = \frac{1}{Z}e^{-\beta\langle\psi|H|\psi\rangle} = \frac{1}{Z}\langle\psi|e^{-\beta H}|\psi\rangle = \frac{1}{Z}\langle\psi|\widehat{U}(\beta)|\psi\rangle, \tag{20}$$

where $Z$ is the partition function, $Z = \sum_{|\psi\rangle} e^{-\beta\langle\psi|H|\psi\rangle}$. When the system is in thermal equilibrium and the temperature is low, ignoring the degeneracy of the ground state, the probability of the ground state is typically close to 1.

To facilitate the calculation of the imaginary time evolution $\widehat{U}(\beta)|\psi\rangle$, the Trotter-Suzuki decomposition is commonly employed. This decomposes the imaginary time evolution operator into the product of local operators. In doing so, we map the original problem Hamiltonian into a classical system consisting of $\tau$ coupled copies of the original system, known as Trotter slices. Taking the Hamiltonian of the Ising model as an example, after mapping, it is

$$H(t) = -\sum_{k=1}^{\tau}(\sum_{(i,j)} J_{i}j\sigma_i^k\sigma_j^k + J(t)\sum_{i=1}^{N}\sigma_i^k\sigma_i^{k+1}), \tag{21}$$

where $\tau$ is the number of Trotter slices, $\sigma_i^k$ is the $i$-th spin on the $k$-th slice, $J(t) = -\frac{\tau T}{2}ln(tanh(\frac{\Gamma(t)}{\tau T}))$ is the coupling function between slices, which is the coupling along the imaginary time dimension, and $J_{ij}$ describes the coupling in the original two-dimensional direction of the Ising model, where $\Gamma(t) = \Gamma_0(t-1)$ is a linear function of time, representing the transverse field strength of the quantum system, time $t \in [0,1]$. On this basis, the Metropolis-Hastings Monte Carlo algorithm is used to explore the solution space. Each exploration is called a Monte Carlo Step (MCS), and each MCS consists of one local flip and one global flip.

- Local movement: A randomly selected spin in a single slice is flipped, and the energy difference is calculated. The change is accepted or rejected using the Metropolis-Hastings criterion, allowing the system to escape from local optima.
- Global flip: All spins of the same qubit across all slices are flipped simultaneously, and the new solution is evaluated based on the energy difference.

SQA reduces system energy by gradually decreasing the transverse field strength. Similar to SA, it explores the solution space through multiple MCS at each transverse field strength. When $\Gamma$ decreases from $\Gamma_0$ to $\Gamma_{final}$ or the system energy reaches 0, annealing is completed. The slice with the lowest energy is then selected as the result. In the early stages, even if a local optimum is surrounded by a potential barrier, the system still has a probability of tunneling into a better solution space, rather than gradually climbing over the energy barrier like in SA. As time progresses, the transverse field strength decreases, and quantum tunneling behavior gradually fades, causing the system's behavior to transition from quantum tunneling to classical optimization, eventually stabilizing at a (possibly local) optimal solution. Although SQA has strong capabilities in avoiding local optima, its efficiency is limited by the simulation method and computational cost. For certain problems, the tunneling rate may be very small, especially when the energy barrier is wide and high, making the algorithm require a long time. Additionally, tunneling may lead to the selection of non-optimal solutions if the optimal solution is surrounded by regions with complex barrier structures.

Table A.1: Experimental results of the RBM-based SQA algorithm on various isomorphic graph instances. The last column, "Time" represents the average time per iteration.

| N | Encoding Qubits | Space Size | Final Hit Rate | Time (s) |
|---|---|---|---|---|
| 4 | 6 | $2^6$ | 1 | 0.00438 |
| | 8 | $2^8$ | 1 | 0.00487 |
| | 16 | $2^{16}$ | 1 | 0.00828 |
| | 16 | $2^{16}$ | 1 | 0.01013 |
| 5 | 9 | $2^9$ | 1 | 0.00471 |
| | 9 | $2^9$ | 1 | 0.00512 |
| | 25 | $2^{25}$ | 1 | 0.02736 |
| | 25 | $2^{25}$ | 1 | 0.04405 |
| 6 | 12 | $2^{12}$ | 1 | 0.00550 |
| | 12 | $2^{12}$ | 1 | 0.00614 |
| | 14 | $2^{14}$ | 1 | 0.00777 |
| | 18 | $2^{18}$ | 1 | 0.01945 |
| 7 | 19 | $2^{19}$ | 1 | 0.01227 |
| | 21 | $2^{21}$ | 1 | 0.01668 |
| | 25 | $2^{25}$ | 1 | 0.02281 |
| | 37 | $2^{37}$ | 1 | 0.11903 |
| 8 | 16 | $2^{16}$ | 1 | 0.01406 |
| | 32 | $2^{32}$ | 1 | 0.04284 |
| | 38 | $2^{38}$ | 1 | 0.07155 |
| | 40 | $2^{40}$ | 1 | 0.07658 |
| 9 | 33 | $2^{33}$ | 1 | 0.04150 |
| | 45 | $2^{45}$ | 1 | 0.10126 |
| | 53 | $2^{53}$ | 1 | 0.18448 |
| | 81 | $2^{81}$ | 1 | 0.53516 |
| 10 | 34 | $2^{34}$ | 1 | 0.04739 |
| | 44 | $2^{44}$ | 1 | 0.08890 |
| | 50 | $2^{50}$ | 1 | 0.13730 |
| | 52 | $2^{52}$ | 1 | 0.23352 |

Table A.2:   The average of the final minimum energy expectations of 100 experiments for SA, PIMC-SQA, and RBM-SQA algorithms on various non-isomorphic graphs.

| N | Space Size | SA | PIMC-SQA | RBM-SQA |
|---|---|---|---|---|
| 6 | $2^{10}$ | 5.236 | 4.064 | 4.000 |
| 6 | $2^{14}$ | 5.188 | 4.184 | 4.000 |
| 6 | $2^{18}$ | 4.824 | 4.090 | 4.000 |
| 6 | $2^{12}$ | 6.254 | 6.004 | 6.000 |
| 7 | $2^{29}$ | 5.410 | 4.198 | 4.000 |
| 7 | $2^{26}$ | 5.292 | 4.202 | 4.000 |
| 7 | $2^{15}$ | 6.090 | 4.278 | 4.000 |
| 7 | $2^{17}$ | 6.754 | 6.054 | 6.000 |
| 8 | $2^{32}$ | 5.946 | 4.380 | 4.000 |
| 8 | $2^{40}$ | 5.782 | 4.420 | 4.000 |
| 8 | $2^{16}$ | 5.058 | 4.112 | 4.000 |
| 8 | $2^{16}$ | 5.056 | 4.120 | 4.000 |
| 9 | $2^{17}$ | 9.416 | 8.358 | 8.000 |
| 9 | $2^{33}$ | 5.630 | 4.114 | 4.000 |
| 9 | $2^{41}$ | 6.060 | 4.375 | 4.000 |
| 9 | $2^{21}$ | 5.216 | 4.168 | 4.000 |
| 10 | $2^{28}$ | 7.900 | 6.142 | 6.000 |
| 10 | $2^{44}$ | 6.148 | 4.540 | 4.000 |
| 10 | $2^{100}$ | 10.506 | 9.706 | 8.000 |
| 10 | $2^{52}$ | 10.051 | 8.780 | 8.000 |
| 10 | $2^{44}$ | 7.523 | 6.113 | 6.000 |
| 11 | $2^{35}$ | 5.682 | 4.182 | 4.000 |
| 11 | $2^{25}$ | 7.312 | 6.080 | 6.000 |
| 11 | $2^{25}$ | 5.446 | 4.082 | 4.000 |
| 11 | $2^{53}$ | 5.474 | 4.102 | 4.000 |
| 12 | $2^{144}$ | 6.336 | 5.074 | 4.000 |
| 12 | $2^{80}$ | 8.925 | 6.606 | 6.000 |
| 12 | $2^{80}$ | 5.678 | 4.124 | 4.000 |
| 12 | $2^{42}$ | 6.908 | 4.448 | 4.000 |
| 13 | $2^{47}$ | 7.455 | 4.618 | 4.000 |
| 16 | $2^{256}$ | 9.306 | 8.740 | 8.000 |
| 17 | $2^{169}$ | 10.584 | 8.762 | 8.000 |
| 18 | $2^{212}$ | 5.987 | 4.290 | 4.000 |

