# OpenReview forum: "RBM-Based Simulated Quantum Annealing for Graph Isomorphism Problems"
_ICLR.cc/2026/Conference — Submitted to ICLR 2026_

### Official Review · Reviewer_JYJM · 2025-10-30

**Soundness:** 2
**Presentation:** 2
**Contribution:** 2
**Rating:** 2
**Confidence:** 4

**Summary:**

This paper proposes a quantum-inspired algorithm for the graph isomorphism problem by combining simulated quantum annealing with neural-network quantum states (NQS). Instead of using traditional Path-Integral Monte Carlo-based simulated quantum annealing, the method uses a Restricted Boltzmann Machine (RBM) to parameterize a variational neural quantum state, enabling efficient sampling. The graph isomorphism problem is formulated as a QUBO/Ising model, and the RBM-based ansatz is optimized with variational Monte Carlo.

**Strengths:**

The paper explores the potential of quantum-inspired techniques for addressing a classical combinatorial optimization problem (e.g., graph isomorphism). It is valuable for advancing the intersection of quantum computing and classical algorithm design. In addition, the paper is clearly written, logically progressing from the established QUBO and Ising formulations of the graph isomorphism problem to the specific implementation details of the RBM ansatz and the VMC optimization process.

**Weaknesses:**

1. Lack of novelty: The application of NQS and RBM-based VMC to find the ground state of quantum many-body systems is a well-established method [Carleo & Troyer (2017)]. The proposed method is a relatively straightforward application of this existing method to the Ising formulation of the graph isomorphism problem. The authors should provide deeper theoretical justification or analysis clarifying why and under what conditions the NQS+VMC framework offers advantages for graph isomorphism.

2. Lack of theoretical guarantee: The proposed method is a heuristic algorithm. The paper provides no formal proof of convergence or analysis of its performance.

3. No comparison to state-of-the-art: The performance evaluation is limited to general-purpose heuristics (classical SA and PIMC-SQA). It does not benchmark against state-of-the-art, specialized classical solvers for graph isomorphism, such as Nauty, which can handle graphs with thousands of vertices. Additionally, the experiments are limited to relatively small graphs (≤10 vertices). Evaluating the method on larger graphs is necessary to demonstrate its practical significance and scalability.

**Questions:**

Same as the weaknesses.

---

### Official Review · Reviewer_oc8i · 2025-10-31

**Soundness:** 3
**Presentation:** 3
**Contribution:** 4
**Rating:** 6
**Confidence:** 3

**Summary:**

This paper introduces quantum technology principles as a method for solving the graph isomorphism problem, which involves determining whether two graphs are structurally identical.
In the paper, the answer space to the graph isomorphism problem is mapped to a quantum many-body system, and the wave function of this quantum system is approximated (NQS) using a constrained Boltzmann Machine (RBM) artificial neural network.

The proposed technique achieves higher hit rates than traditional SA and conventional methods, and shows much faster computational speed.
It is an evident and logically well-written paper that requires a high level of expertise, which can be somewhat unfriendly to non-majors.

**Strengths:**

The methodology proposed in this paper is based on standard pipelines that have already been validated in the fields of computational physics and machine learning, and thus has high technical feasibility. Moreover, the key arguments are supported by strong experimental evidence, clearly demonstrating the superiority of the proposed method over SA and PIMC-SQA in terms of accuracy and computational speed.


This paper deals with the problem of graph isomorphism (GI), a fundamental challenge in computer science. The main contribution of this paper is to show that conventional complex quantum simulation schemes (PIMC-SQA) are inefficient, and to demonstrate that variational approaches (VMCs) using pure classical software (RBM neural networks) can be faster and more accurate. This shows that instead of relying on noisy quantum hardware or inefficient simulations in the NISQ era, efficient exploitation of quantum principles with classical machine learning techniques can be a more practical and robust solution.

**Weaknesses:**

This paper assumes significant prior knowledge of both quantum physics (Hamiltonian, Ising model, wave function, NQS, VMC, Trotter decomposition) and machine learning (RBM, variational principle).
The core methodology, Section 3, is mathematically very dense. In particular, the process by which QUBO leads to Hamiltonian and back to the RBM wave function (NQS) can be difficult to follow unless you are an expert in the field.


If the 'Trotter-free' approach is the critical reason for the speedup, this difference needs to be explicitly emphasized and explained in the text. The present description makes it difficult to understand why the RBM-SQA is faster intuitively.


The experiment was conducted only on a relatively small graph with N=18. The complexity of the GI problem increases exponentially with N, making it challenging to determine whether this approach will be effective at larger scales, despite success stories up to N=18.

**Questions:**

Authors' comments on the scalability of the proposed approach compared to SA/PIMC-SQA are needed.


Is it correct to understand that the key reason RBM-SQA is 63.5% faster than PIMC-SQA is that it does not use Trotter decomposition?
And how this benefit scales as the number of qubits increases (asymptotic scaling) needs to be explained.

---

### Official Review · Reviewer_KfW4 · 2025-11-01

**Soundness:** 2
**Presentation:** 2
**Contribution:** 1
**Rating:** 2
**Confidence:** 5

**Summary:**

The submission proposes using RBM-SQA, an annealing method with a neural network ansatz as the variational state and a graph-encoded Hamiltonian, to solve QUBO problems reformulated from graph isomorphism (GI) problems. The draft focuses on experimental validation, comparing results to two different annealing methods. The results show that RBM-SQA can solve the given GI problems efficiently.

**Strengths:**

1. The authors demonstrate a pipeline to use variational neural network state for GI problems
2. Experiments shows good results while keep efficiency comparing to another annealing methods

**Weaknesses:**

1. This approach is not novel. Prior work (arxiv:2101.10154; 2002.02973 and others) has already demonstrated neural network quantum states for annealing and related tasks. The manuscript needs to clarify its unique contribution.
2. The paper lacks clarity; the discussion of encoding methods from GI problems to QUBO is unclear.
3. The experimental validation is weak.
    1. It does not compare to the WL test, which is the standard baseline for assessing graph isomorphism distinguishability.
    2. The paper does not report scalability analysis, runtime statistics, or success probability on structured benchmark families such as strongly regular graphs or CFI graphs, cases where WL tests are known to fail.
4. There are several typographical errors and inconsistent notation throughout.

**Questions:**

1. The approach is fully classical; why do the authors claim "quantum tunneling" or other quantum effects?
2. Could the authors distinguish more clearly between variational optimization of an RBM energy model and genuine quantum-annealing dynamics?
3. Can the authors justify their choice of RBM over other models?

---

### Meta-Review · Area_Chair_5NwY · 2026-01-06

**Summary:**

The paper proposes an RBM-based Simulated Quantum Annealing (SQA) approach to solve the Graph Isomorphism (GI) problem by formulating it as a QUBO optimization task. While the reviewers acknowledged the technical validity of connecting Neural Quantum States with annealing, the consensus is to reject the submission due to limited novelty and insufficient experimental validation. The experiments were restricted to trivially small graphs ($N \le 18$) and failed to include necessary comparisons against standard, highly efficient classical GI solvers (e.g., Nauty, Bliss) or the Weisfeiler-Lehman test, making it impossible to assess the method's practical utility.

**Reviewer Concerns:**

**Addressed:**

None of the major concerns were effectively resolved during the rebuttal phase.

**Outstanding:**

*   **Baselines and State-of-the-Art:** Reviewers **KfW4** and **JYJM** strongly emphasized the lack of comparison against specialized classical solvers for GI (like Nauty or Bliss) or the Weisfeiler-Lehman (WL) test. Comparing only against other general-purpose annealing heuristics is insufficient for GI, where specialized algorithms are exponentially faster for the graph sizes tested.
*   **Scalability and Experimental Scope:** All reviewers, including **oc8i**, noted that experiments on graphs with $N \le 18$ vertices are unconvincing. The authors failed to provide the requested asymptotic scaling analysis to demonstrate utility beyond toy problems.
*   **Novelty:** Reviewers **KfW4** and **JYJM** pointed out that using RBMs for Variational Monte Carlo is a well-established technique and mapping GI to QUBO is standard; thus, the work lacks distinct algorithmic contribution.

**Reviewer Scores:**

**Reviewer KfW4: 2**
The rebuttal failed to provide the theoretical justification or large-scale experiments (such as comparisons to the WL test) required to change the assessment.

**Reviewer oc8i: 4**
While initially appreciating the technical feasibility, the reviewer acknowledged that the small scale ($N=18$) is a significant weakness and agreed that the absence of standard baselines lowers the paper's significance.

**Reviewer JYJM: 2**
The score remains low due to the fundamental lack of comparison to state-of-the-art specialized solvers (e.g., Nauty) and the absence of theoretical guarantees for the proposed heuristic.

---

### Decision · Program_Chairs · 2026-01-26

Reject